# Effects of Jet Milling on the Physicochemical Properties of Buckwheat Flour and the Quality Characteristics of Extruded Whole Buckwheat Noodles

**DOI:** 10.3390/foods11182722

**Published:** 2022-09-06

**Authors:** Jiayu Cheng, Sijia Lei, Li Gao, Yingquan Zhang, Weiwei Cheng, Zhenjiong Wang, Xiaozhi Tang

**Affiliations:** 1Key Laboratory of Grains and Oils Quality Control and Processing, Collaborative Innovation Center for Modern Grain Circulation and Safety, College of Food Science and Engineering, Nanjing University of Finance and Economics, Nanjing 210023, China; 2Key Laboratory of Agro-Products Processing, Ministry of Agriculture and Rural Affairs, Institute of Food Science and Technology, Chinese Academy of Agricultural Sciences, Beijing 100193, China

**Keywords:** jet milling, particle size, damaged starch, extruded whole buckwheat noodles

## Abstract

The effects of jet milling on the physicochemical properties of buckwheat flour and the quality characteristics of extruded whole buckwheat noodles (WBN) were investigated in this study. The results reveal that the application of jet milling significantly reduced the particle size of buckwheat flour. As a result, the damaged starch content, water solubility index, water absorption index and swelling power of buckwheat flour all increased. It was worth noting that moderately ground buckwheat flour powder (D_50_ = 65.86 μm) had the highest pasting viscosity and gel hardness. The breaking rate and cooking loss of extruded whole buckwheat noodles made from the above powder were reduced by 33% and 16%, respectively. Meanwhile, they possessed the highest lightness and firmest network structure. Jet milling increased the soluble dietary fiber (SDF) content from 3.45% to 4.39%, and SDF further increased to 5.28% after noodle extrusion. This study was expected to provide a reference for exploiting high-quality gluten-free noodles from the perspective of milling.

## 1. Introduction

Buckwheat has high nutritional value, with a higher content of dietary fiber, vitamins, phosphorus, calcium, iron, lysine, linoleic acid, niacin and rutin. It also has high medicinal value, exhibiting hypocholesterolemic, antihypertensive, anticancer, anti-inflammatory, antidiabetic, and neuroprotective activity [1]. Therefore, buckwheat has great potential as a staple food for different types of consumers [2]. 

The remarkable health benefits of buckwheat products contribute to their widespread popularity in most East Asian countries and parts of European countries [3]. However, the development of whole buckwheat noodles has been limited so far, because of the high breaking rate and cooking loss. The protein composition of buckwheat is different from that of wheat, and so it struggles to form effective protein networks like those seen in a gluten network [4]. The addition of wheat flour or exogenous gluten is usually used to compensate for this deficiency. It was reported that buckwheat flour could not be used in large quantities (basically 30% or less) during the conventional noodle processing process, and 20% addition showed the most acceptable sensory qualities [2]. Extrusion processing is a physical technology which is characterized by no pollution, high efficiency and energy savings. During extrusion, starch granules are gelatinized and partly degraded by heat and shear effects. The starch network is well-formed, replacing the protein network to some extent [5]. In our previous studies, extrusion processing instead of conventional noodle processing techniques was successfully utilized to produce whole buckwheat noodles [6].

Recently, the development and application of fine or superfine grinding, one of the typical techniques of which is jet milling, in whole grain-based food research have been widely studied [7]. The decrease in the particle size of materials to a micro size, which causes some changes in physicochemical properties, such as structure and surface area, and brings about some unexpected characteristics. The physical and processing properties of whole grain flours can be improved by fine or superfine grinding, thus leading to a higher quality of the final products. For example, Wang et al. [8] reported that the decrease in whole wheat flour particle size obviously enhanced the tensile resistance and extensibility of dough. Niu et al. [9] also showed that milled materials with finer particle sizes presented positive effects on the quality improvement of whole wheat noodles.

Hence, the purposes of this research were to investigate the effects of jet milling on the physicochemical properties of buckwheat flours and the cooking quality, textural properties, and proximate compositions of extruded whole buckwheat noodles. This study is expected to provide meaningful information on exploiting whole buckwheat noodle production for commercial practice.

## 2. Materials and Methods

### 2.1. Materials

Dehulled buckwheat grain (*Fagopyrum esculentum Moench*) was bought from Yanzhifang Food Co., Ltd. (Anhui, China). The cultivar of buckwheat above is widely grown in Liaoning, China. The grains were ground by an ultra-speed centrifugal pulverizer (ZM 200, Retsch, Haan, Germany), of which the electrode speed was set to 14000 rpm, and passed through a 0.5 mm sieve to obtain coarse buckwheat flour (BF1). The chemical components of BF1 (dry basis) were analyzed according to the AACC standard methods 46-11.02, 30-25.01, 08-01.01 and 76-13.01 (AACC, 2000) [10], of which the protein content was 15.07 ± 0.07%, that of crude fat was 2.57 ± 0.03%, that of total ash was 2.26 ± 0.03%, and the total starch content was 75.85 ± 0.32%. The Total Dietary Fiber Kit was bought from Megazyme International Ireland Ltd. (Wicklow, Ireland). All other chemical reagents utilized in this research were of analytical grade

### 2.2. Preparation of Buckwheat Fine Powder by Jet Milling

The resulting coarse buckwheat flour was pulverized by means of J-50 jet milling (TECNOLOGIA MECCANICA, Bergamo, Italy). The crushing pressure was 12 MPa, and the feeding pressure was 10 MPa. Three treatments with different feeding speeds of 145 r/min, 200 r/min and 250 r/min were applied to obtain fine buckwheat flour with varying particle sizes, defined as BF2, BF3 and BF4, respectively. 

### 2.3. Particle Size 

The particle size distribution of samples was tested in wet method mode using a S3500 Particle Size Analyzer (Microtrac Co., Ltd., Montgomeryville, PA, USA). Referring to Yu et al. [11], ultra-pure water was used as a dispersant, and the refractive index of the samples was set to 1.434. The data were analyzed by the system software FLEX 10.5.3.

### 2.4. Damaged Starch (DS)

Damaged starch (DS) of buckwheat flour samples was measured according to the AACC Method 76-30A (AACC, 2010) [12]. A 1.0 g sample and 50 mg of alpha-amylase were added into 45 mL of acetate buffer, and incubated at 30 °C for 15 min. The above suspension was mixed evenly with 3 mL of H_2_SO_4_ solution and 2 mL of sodium tungstate solution, and then filtered. The results were quantitated by titrating to measure reducing sugars.

### 2.5. Color 

Color analysis (L*, a* and b*) was performed by using a CM-5 chroma meter (Konica Minolta, Osaka, Japan). A white standard plate was used for calibration, and then the samples were placed and measured in a sample holder. The data were expressed as the mean of 10 measurements taken at random locations in the sample.

### 2.6. Hydration Properties 

The hydration properties of buckwheat flour were measured on the basis of the method of Anderson et al. [13] with minor adjustment. A 1.0 g sample (W_0_, dry basis) was put into a centrifuge tube (W_1_) filled with 25 mL of distilled water and vibrated to disperse it completely. The dispersion was incubated in a water bath at 100 °C for 30 min and shaken for 30 s every 10 min, followed by centrifugation at 4200 r/min for 15 min. The supernatant was decanted into a pre-weighed 500 mL beaker (W_2_) and drained at 105 °C until reaching the constant weight (W_3_). The dry residue in the centrifugal tube was weighed (W_4_). The water solubility index (WSI), water absorption index (WAI) and swelling power (SP) were calculated as follows:(1)$$\mathrm{WAI}=\frac{W_{4}-W_{1}}{W_{0}}\times 100\%$$
(2)$$\mathrm{WSI}=\frac{W_{3}-W_{2}}{W_{0}}\times 100\%$$
(3)$$\mathrm{SP}=\frac{W_{4}-W_{1}}{W_{0}\times \left(1-\mathrm{WSI}\right)}$$

### 2.7. Pasting Properties

Pasting properties were tested by a Rapid Visco Analyser (RVA, Model Super-3, Newport Scientific, Warriewood, NSW, Australia), according to the AACC Method 76-21.01 (AACC, 2000) [14]. The values of peak viscosity, trough viscosity, breakdown, final viscosity, and setback were calculated.

### 2.8. Rheological Properties

The paste samples were obtained after the RVA measurement, then transferred to the rheological test by a dynamic rheometer (MCR 301, Anton Paar, Graz, Austria). The measurement parameters were as follows: plate diameter, 50 mm (rotor: PP50); strain, 0.5%; temperature, 25 °C; frequency, 0.1–20 Hz; and gap, 1 mm. The storage modulus (G′), loss modulus (G″) and tan δ were recorded.

### 2.9. Gel texture Properties

The paste samples by RVA were put into a sealed plastic mold (the sample size was 2 cm × 2 cm × 2 cm cube) and then held at 4 °C for 24 h to form a stable gel system. The textural properties of samples were determined using a texture analyzer (TA-XT 2i Stable Micro Systems, Surrey, UK). The texture profile analysis (TPA) mode was selected and the test conditions were as follows: pretest speed, 5.00 mm/s; test and posttest speed, 2.00 mm/s; strain, 65%; trigger force, 5.0 g; interval time, 2 s and data acquisition rate, 200 PPS. Hardness, springiness, cohesiveness, chewiness and resilience were calculated. Six replicates were performed.

### 2.10. Preparation of Extruded Whole Buckwheat Noodles (WBN)

Based on our previous work [5,6], extrusion runs were performed by a DSE-20/40D twin screw extruder (Bradender, Germany), with a screw length to diameter ratio (L/D) of 40:1 and a die size of 2 mm. The screw of the extruder was divided into 6 zones, with temperatures set at 40:60:110:90:80:80 °C from zone 1 to zone 6. The screw speed was set constant at 120 rpm and the moisture content was adjusted to 38%. The extruded whole buckwheat noodles prepared by BF1, BF2, BF3 and BF4 were defined as WBN1, WBN2, WBN3 and WBN4, respectively.

### 2.11. Scanning Electron Microscopy (SEM)

Buckwheat flours (BF), WBN and cooked WBN at the optimal cooking time were freeze-dried for further SEM observation. Detailed SEM test methods referred to Liu et al. [15].

### 2.12. Cooking Properties

The cooking properties of WBN were determined based on AACC Method 66-50 (AACC, 2000) [16] and were described in detail in our previous work [6].

### 2.13. Texture Properties of WBN

The texture properties of WBN were analyzed according to Liu et al. [11] with slight modifications. The textural profile analysis (TPA) mode was applied by a TA-XT2i Texture Analyser (Stable MicroSystems, Surrey, UK). The noodles were cooked at the optimal cooking time and the measurements were carried out with six replicates. The test conditions were as follows: cylindrical probe (P/36R); pretest speed, 5.00 mm/sec; test and posttest speed, 2.00 mm/s; strain, 75%; trigger force, 5.0 g; interval time, 5 s; and data acquisition rate, 400 PPS.

### 2.14. Determination of Dietary Fiber Content

Referring to AOAC Method 991.43 (AOAC, 1999) [17], the insoluble dietary fiber (IDF), soluble dietary fiber (SDF) and total dietary fiber (TDF) in flour (BF) before extrusion and in noodles (WBN) after extrusion were tested by a FOSS fiber analyzer (Fibertec 8000, Hoganas, Sweden) with Megazyme K-TDFR kit.

### 2.15. Statistical Analysis

All measurements were carried out in triplicate unless mentioned specifically. Data analysis was accomplished using the statistical software Origin 8.0 (OriginLab, Northampton, MA, USA) and SPSS 18.0 (IBM, USA). ANOVA and Duncan tests were utilized to assess statistical differences between the mean values (*p* < 0.05).

## 3. Results and Discussion

### 3.1. Particle Size and Damaged Starch

The particle size of cereal flour has an obvious effect on the processing characteristics of related products, which is deemed as a vital index to judge product performance. The particle size distribution and damaged starch of various buckwheat flour samples are presented in Table 1. The results indicate that a different particle size distribution of buckwheat flour can be obtained by controlling the feed speed of jet milling. The average particle size (D_50_) of coarse buckwheat flour was 82.27 μm. After jet milling treatment, D_50_ gradually decreased from 65.86 μm to 20.57 μm with the increase in feed speed from 145 r/min to 250 r/min. In addition, SEM images (Figure 1) reflected that with the increase in feeding speed, the particle size of buckwheat flour gradually decreased, and the uniformity of particle size distribution increased.

Damaged starch is a recognized and crucial criterion of flour product quality, which is closely related to many important properties, such as hydration and pasting characteristics [11,18]. As shown in Table 1, damaged starch content was negatively related to average particle size. Along with the reduction in particle size, the damaged starch content significantly increased in buckwheat flour from 11.45% of BF1 to 21.43% of BF4 as they were exposed to more severe mechanical effects.

### 3.2. Hydration Properties

The influence of particle size on the hydration properties of buckwheat flours is depicted in Table 1. Compared with BF1, the water absorption index (WAI), water solubility index (WSI) and swelling power (SP) of BF2, BF3 and BF4 gradually increased as the particle size of buckwheat flour decreased. The WAI of buckwheat flour increased from 7.01 g/g to 7.65 g/g, the WSI of buckwheat flour increased from 0.16 g/g to 0.18 g/g, and the SP of buckwheat flour increased from 4.02 g/g to 4.85 g/g.

Previous research also reported the negative correlations between particle size and flour hydration properties [18,19]. The possible reason for the increase in WAI was that a smaller particle size gave a greater surface area, which resulted in stronger water affinities. Flour particles with smaller sizes also need a shorter time to permeate inside [19]. The higher WSI value was probably associated with the damaged starch content. In addition, the jet milling technology increased the content of soluble fiber in buckwheat flour, which led to greater loss of material to water [18]. The swelling power is the ability of materials (usually insoluble starch or flour granules) to retain water molecules. The increase in swelling power might be because of the changes in starch structures caused by jet milling, which brought about the exposure of crystalline structures of starch to water [20].

### 3.3. Pasting Properties

The pasting characteristics of buckwheat flours are shown in Table 2. As can be observed from Table 2, jet milling had a significant influence on the pasting properties of buckwheat flour. Compared with BF1, BF2 had higher peak viscosity, final viscosity, trough viscosity and setback value, and lower breakdown value. This might be related to the fact that the pasting properties of buckwheat flour was not only connected to the gelatinization of starch but also related to the gelation of oligosaccharides and some other polysaccharides in buckwheat flour. Proper grinding facilitated the slight degradation of starch and the release of non-starch saccharides from fibers, thus resulting in enhanced pasting viscosities [9]. What is more, since finer powder meant a greater damaged starch content in this study, damaged starch hydrated better, leading to higher pasting viscosities, within a certain range [21]. The decrease in breakdown value and the increase in setback value indicated that the stability of hot paste increased, but the stability of cold paste decreased. Proper retrogradation was beneficial to produce starch-based gel products. However, with the further decrease in particle size, the pasting viscosity (i.e., peak viscosity, trough viscosity and final viscosity) of BF3 and BF4 gradually decreased. This could be attributed to the higher mechanical shear force and the more severe damage to the starch. The grinding treatments could lead to the physical destruction of the starch granules, including the reduced crystalline region and degradation of starch molecules. The change was closely related to starch swelling and gelling behaviors and thus gave rise to the changes in starch pasting properties [20]. The existence of excessive damaged starch might have a diluting and hindering effect on the starch gel matrix, thus reducing the pasting viscosity [22].

### 3.4. Rheological Properties

The dynamic viscoelastic characteristics of buckwheat paste were studied as a function of frequency (Figure 2). G′ and G″ represent the elastic and viscous attributes of samples, respectively, and tan δ indicates the ratio of viscous to elastic components [23]. From Figure 2, the following results were obtained. G′ for all the samples was higher than G″, and the tan δ values were lower than 1 over the frequency range, showing their weak viscoelastic gel behaviors. Compared with BF1, with the decrease in buckwheat flour particle size, G′ and G″ first increased for BF2 and then decreased for BF3 and BF4. At the same time, tan δ first decreased for BF2 and then increased for BF3 and BF4. The results suggest that a reduction in particle size enhanced the elasticity, strength, and rigidity of the gel system, which was beneficial for gluten-free pasta products. However, with the further decrease in particle size, the gel properties were weakened due to the rapid rise in the amount of damaged starch. The overall trend of G′ and G″ was positively related to the trend of RVA pasting viscosity.

### 3.5. Texture Profile Analysis of Gels

The effect of different particle sizes on textural parameters of buckwheat gels is presented in Table 3. Compared with BF1, the gel hardness, springing and chewiness value of buckwheat flour went up and then down with decreasing particle size, exhibiting a similar trend to the final viscosity in RVA and G′ and G″ in rheological curves. Similar results were also found by Wang et al. [24], who investigated the effect of particle size on the quality attributes of reconstituted whole wheat flour. The gel hardness of flour, which is predominantly affected by the content of amylose, the molecular structure of amylopectin, as well as the interactions between dispersed and continuous phases of the gel, is considered to be a dominant indicator influencing the texture of cooked starch noodles [25]. BF2 has a higher gel hardness, springiness and chewiness of 25.71 g, 0.90 and 12.30 g, respectively. The increase in gel hardness after jet milling for BF2 was probably attributed to the slight degradation of starch and the release of non-starch saccharides, which could help to form a continuous and stronger gel network structure. However, when the particle size was further decreased to BF3 and BF4, gel hardness, springiness and chewiness decreased due to the much higher content of damaged starch, which weakened the gel network.

### 3.6. Color Analysis

The results of color measurement of buckwheat flours (BF) and WBN are presented in Table 4. Product color is considered as one of the key factors in determining consumer acceptance [26]. Compared with BF1, the lightness value (L*) of jet-milled flours increased from 88.02 to 89.51, the redness value (a*) increased from 0.67 to 0.76, and the yellowness value (b*) decreased from 9.18 to 8.76 due to the reduction in particle size. Niu et al. [9] investigated the effects of superfine technology on the color of whole wheat flour, and obtained a similar finding regarding the increase in L* following the reduction in particle size. The reason for this might be that the smaller the particle size was, the heavier the mechanical shearing force applied during the grinding process. The higher shear force might induce degradation of the pigment in buckwheat flour, leading to a higher value of the lightness of powder. Additionally, for the granules with a smaller particle size, its specific surface area was larger, leading to a higher reflective rate.

In addition, compared to BF, WBN possessed lower L* as well as higher a* and b*, which indicated that the extrusion process would darken the flour products. The pigment oxidation and Maillard reactions promoted these changes [27]. Interestingly, the trend of chromaticity values between BF and WBN was not consistent. WBN2 and WBN4 showed the highest L*, which might be in line with consumers’ preferences. Within limits, finer powders implied greater specific surface area and greater probability of chemical reactions between related components, further promoting the pigments oxidation and Maillard reactions [28].

### 3.7. Effects of Jet Milling and Extrusion on Dietary Fiber Content

Figure 3 shows that before extrusion, jet milling visibly enhanced SDF content from 3.45% to 4.39% and reduced IDF content from 6.61% to 5.69%; nevertheless, TDF was almost unchanged compared to BF1, illustrating that jet milling gave rise to a redistribution of fiber components in TDF. Similar conclusions were drawn by Zhu et al. [29]. The increase in soluble dietary fiber is beneficial for improving the hydration properties of flour.

It is interesting that SDF further increased to 5.28%, and IDF decreased to 5.11% subsequently after extrusion. The TDF also increased after extrusion. These phenomena are mainly related to the fact that the relatively high temperature, pressure, and strong shear force during the extrusion process induced the breakdown of glycosidic bonds of insoluble polysaccharide molecules and converted them into smaller soluble constituents. In addition, the breaking of chemical bonds in dietary fibers during extrusion also changed the molecular polarity of insoluble dietary fibers [30]. Similar results were reported by Rashid et al. [31], who suggested that the extrusion processing had a beneficial influence on the total and soluble dietary fiber content.

Soluble dietary fiber is reported to be more valuable than insoluble dietary fiber in terms of function and nutrition because of its good preventive effect on obesity and cardiovascular disease [32]. In view of food applications, soluble dietary fiber possesses better taste and texture when compared to insoluble dietary fiber and is more suitable for application in food processing [33]. Thus, the increase in soluble dietary fiber content after jet milling and extrusion is beneficial for the application of buckwheat flour in food processing from the perspective of nutrition and the edible quality of flour products.

### 3.8. Microstructure of WBN

The scanning electron micrographs (SEM) of WBN are presented in Figure 3A–D. It can be clearly seen that the cross-sectional structure of all samples was compact, especially WBN2, followed by WBN3. WBN1 displayed more cracks and laminations.

Since the cross sections of extruded noodles were too dense, in order to further analyze the network structure, extruded buckwheat noodles were cooked for the optimum cooking time for observation (Figure 4E–H). It was obvious that the network matrix of WBN1 had insufficient support force and low network strength, and thus there were many folds and collapses. However, the gel network of WBN2 was firm and the pore walls were thicker, which might be related to the binding role of damaged starch. Meanwhile, excessive damaged starch was detrimental to the gel matrix and gathered together with fiber and other components to form fragments, which existed independently outside the network system, resulting in an incomplete network structure, multiple fractures and many fragments, as shown in Figure 4G–H. These results are in accordance with the gel texture properties discussed above.

### 3.9. Cooking Properties of WBN

Cooking properties, such as water absorption rate, cooking loss, strip broken rate and cooking time, are important parameters for quality evaluation of noodles [34]. The cooking parameters of WBN are shown in Table 5. Among these parameters, the cooking loss of noodles, which is a key indicator for noodle quality evaluation, is a quantitative reflection of the soup turbid degree of noodles in the cooking process. The higher cooking loss rate indicates that more substances are dissolved into the water during the cooking process. As can be observed from Table 5, the cooking loss of WBN first decreased and then increased with the decrease in particle size. The WBN2 exhibited the lowest cooking loss of 7.86. This might be mainly attributed to the better gel forming property of BF2, leading to lower cooking loss of noodles. This result was also in agreement with that of the gel texture and SEM results. For WBN3 and WBN4, excessive grinding could distort the starch crystalline structure and induce amylose leaching out of starch granules, thus resulting in greater cooking loss [9].

The broken rate and water absorption showed the same trend as cooking loss, while the cooking time presented an opposite trend. The increase in the cooking time of WBN2 was probably caused by the formation of a compact texture during extrusion cooking. When the particle size was further decreased, the cooking time exhibited a decreasing trend, probably owing to the more severe damage of starch and fragmentation of gel texture as shown by SEM. With the proper content of damaged starch, water absorption increases, which might help bind the dough and improve the compactness of the noodles. However, if starch damage is excessive, the noodles’ texture tends to be fragmented, inducing higher cooking loss and breaking rate of the cooked noodles [35]. Obviously, the particle size or damaged starch content of flour was not directly or inversely proportional to the cooking quality of products. These results suggest that the moderate jet milling treatment could significantly improve the cooking quality of extruded whole buckwheat noodles.

### 3.10. Effects of Jet Milling on the Textural Properties of WBN

The textural characteristics of noodle products play a crucial role in consumer acceptance. What is more, hardness is the most important index of cooked noodles’ quality. Noodle textural attributes were revealed in Table 6. As listed in Table 6, the textural parameters of cooked buckwheat noodles were strongly linked with the textural parameters of gel, of which the WBN2 exhibited significantly higher hardness, springiness, cohesiveness, chewiness, and resilience. These results were contributed to by the stronger gel network structure due to the fine particle size and moderate starch damage. Some studies [36,37] have suggested that, to a certain extent, the finer the particles, the better the textural parameters such as hardness and elasticity of the noodles, which was conducive to improving the consumer acceptability of products.

## 4. Conclusions

Particle size and damaged starch content appeared to play an important role in influencing the quality attributes of gluten-free noodles. Hydration characteristics and gelling behavior of buckwheat flour were significantly affected by the feeding speed of jet milling. The physicochemical properties, cooking properties, and microstructure of extruded noodles were improved significantly by the proper grinding process where the feeding speed of jet milling was 145 r/min and the particle size (D_50_) of flour was reduced to 65.86 μm. However, excessive starch damage induced by the jet milling process exerted some adverse effects on the properties of noodles. It is worthwhile to give careful consideration to the particle size and milling degree, which may be crucial to food manufacturers from a processing point of view. Further studies need to be performed on the sensory properties of the extruded whole buckwheat noodles in order to evaluate their consumer acceptability.

## Figures and Tables

**Figure 1 foods-11-02722-f001:**
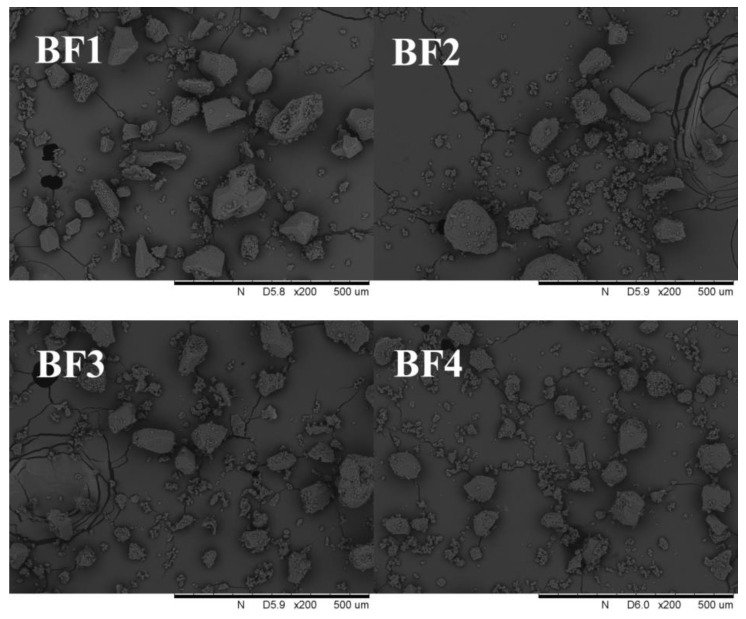
SEM micrographs of buckwheat flour. BF1: coarse buckwheat flour. BF2–4: jet-milled buckwheat flour prepared by adjusting the feeding speed (145 r/min, 200 r/min and 250 r/min), respectively.

**Figure 2 foods-11-02722-f002:**
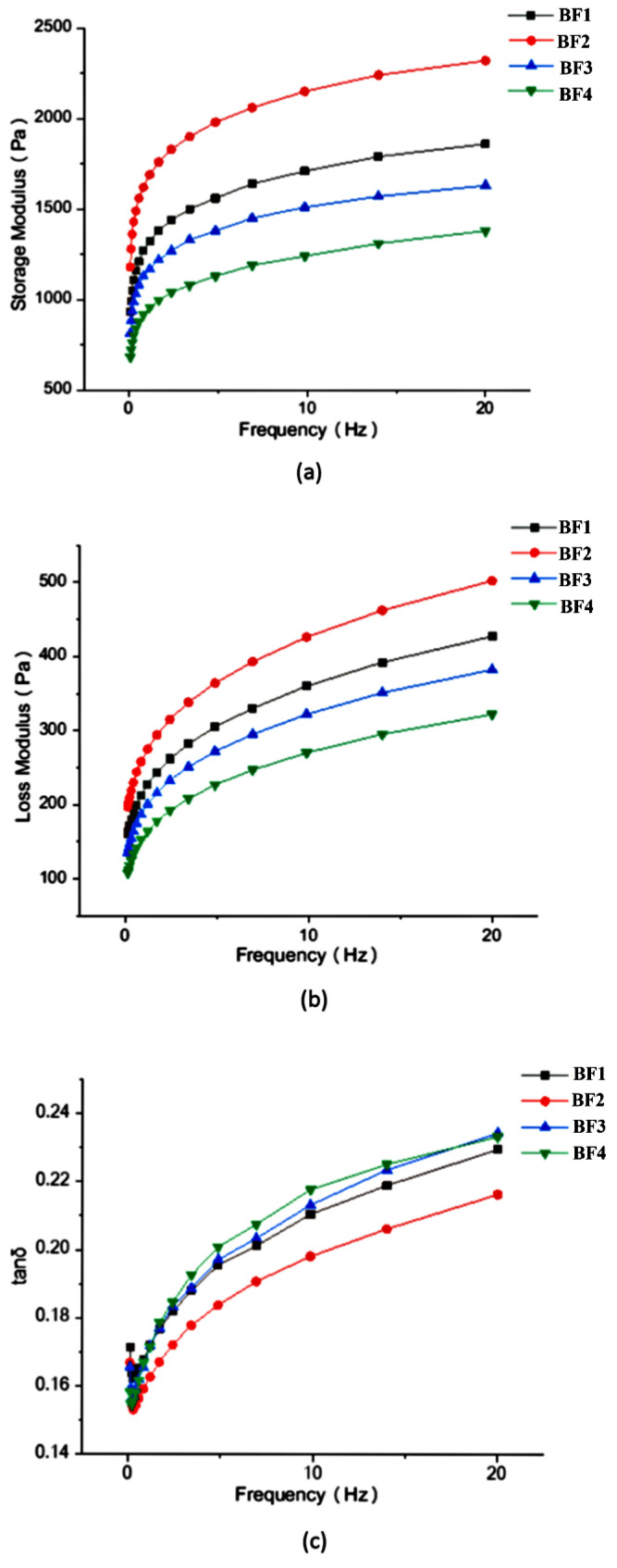
Viscoelastic properties of buckwheat pastes with different particle size: (**a**) storage modulus; (**b**) loss modulus; (**c**) tan δ.

**Figure 3 foods-11-02722-f003:**
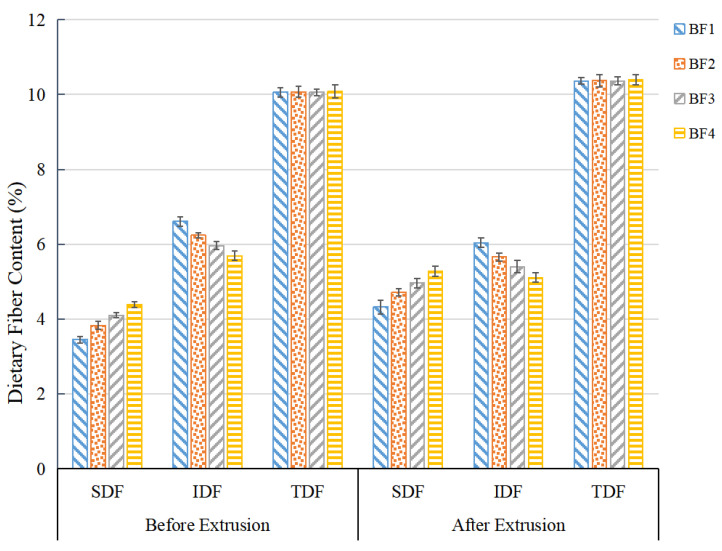
Dietary fiber content of WBN before and after extrusion: SDF, soluble dietary fiber; IDF, insoluble dietary fiber; TDF, total dietary fiber.

**Figure 4 foods-11-02722-f004:**
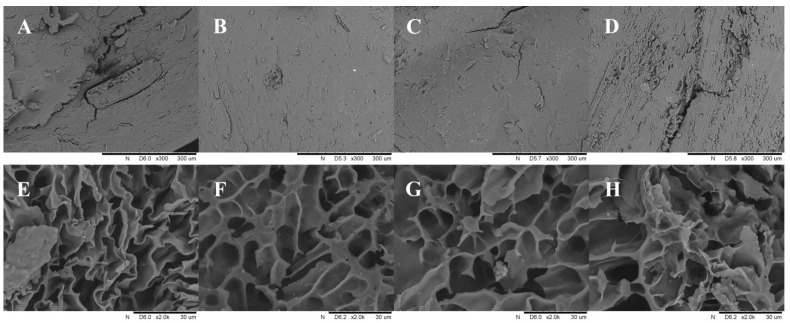
SEM micrographs of WBN: (**A**–**D**) extruded whole buckwheat noodles prepared by BF1–4, respectively; (**E**–**H**) extruded whole buckwheat noodles cooked for the optimal cooking time, respectively.

**Table 1 foods-11-02722-t001:** The analysis of the particle size, damaged starch and hydration properties of buckwheat flour.

Sample	D_10_ (μm)	D_50_ (μm)	D_90_ (μm)	Damaged Starch (%)	WAI (g/g)	WSI (g/g)	SP
BF1	15.93 ± 1.10 ^a^	82.27 ± 2.13 ^a^	198.20 ± 3.11 ^a^	11.45 ± 0.49 ^a^	7.01 ± 0.07 ^bc^	0.16 ± 0.00 ^c^	4.02 ± 0.02 ^cd^
BF2	7.40 ± 0.27 ^b^	65.86 ± 1.23 ^b^	166.20 ± 2.10 ^b^	16.56 ± 0.21 ^b^	7.17 ± 0.01 ^b^	0.16 ± 0.00 ^c^	4.21 ± 0.03 ^c^
BF3	6.87 ± 0.11 ^c^	40.28 ± 1.07 ^c^	145.70 ± 1.74 ^c^	19.04 ± 0.56 ^c^	7.19 ± 0.10 ^b^	0.17 ± 0.00 ^b^	4.62 ± 0.06 ^b^
BF4	5.61 ± 0.14 ^d^	20.57 ± 0.91 ^d^	132.50 ± 1.19 ^d^	21.43 ± 0.40 ^d^	7.65 ± 0.36 ^a^	0.18 ± 0.01 ^a^	4.85 ± 0.16 ^a^

Different small letter superscripts in the same column indicate that the values are significantly different at *p* < 0.05. The results are presented as the mean of the equivalent diameters at cumulative volumes of 10% (D_10_), average particle size (D_50_) and equivalent diameters at cumulative volumes of 90% (D_90_). WAI: water absorption index; WSI: water solubility index; SP: swelling power. BF1: coarse buckwheat flour. BF2-4: jet-milled buckwheat flour prepared by adjusting the feeding speed (145 r/min, 200 r/min and 250 r/min), respectively.

**Table 2 foods-11-02722-t002:** The analysis of the pasting properties of buckwheat flour.

Sample	Peak Viscosity (cp)	Trough (cp)	Breakdown (cp)	Final Viscosity (cp)	Setback (cp)
BF1	2937 ± 25 ^b^	2564 ± 8 ^b^	373 ± 17 ^a^	4987 ± 121 ^b^	2424 ± 114 ^b^
BF2	3054 ± 12 ^a^	2694 ± 74 ^a^	360 ± 79 ^b^	5173 ± 40 ^a^	2479 ± 42 ^a^
BF3	2841 ± 14 ^c^	2511 ± 12 ^c^	318 ± 2 ^d^	4852 ± 32 ^c^	2329 ± 44 ^c^
BF4	2747 ± 28 ^d^	2411 ± 62 ^d^	337 ± 34 ^c^	4582 ± 113 ^d^	2172 ± 52 ^d^

Different small letter superscripts in the same column indicate that the values are significantly different at *p* < 0.05.

**Table 3 foods-11-02722-t003:** The analysis of the textural properties of buckwheat gel.

Sample	Hardness (g)	Springiness	Cohesiveness	Chewiness (g)	Resilience
BF1	19.78 ± 0.82 ^c^	0.70 ± 0.03 ^c^	0.51 ± 0.01 ^c^	7.03 ± 0.09 ^c^	0.01 ± 0.00 ^b^
BF2	25.71 ± 0.59 ^a^	0.90 ± 0.02 ^a^	0.53 ± 0.03 ^a^	12.30 ± 0.13 ^a^	0.02 ± 0.00 ^a^
BF3	20.82 ± 0.75 ^b^	0.82 ± 0.05 ^b^	0.52 ± 0.04 ^b^	8.79 ± 0.48 ^b^	0.01 ± 0.00 ^b^
BF4	19.15 ± 0.49 ^c^	0.71 ± 0.01 ^c^	0.51 ± 0.01 ^c^	6.94 ± 0.03 ^cd^	0.01 ± 0.00 ^b^

Different small letter superscripts in the same column indicate that the values are significantly different at *p* < 0.05.

**Table 4 foods-11-02722-t004:** The analysis of color of buckwheat flour and extruded whole buckwheat noodles (WBN).

Sample	L*	a*	b*
BF1	88.02 ± 0.06 ^b^	0.67 ± 0.02 ^e^	9.18 ± 0.11 ^c^
BF2	88.89 ± 0.26 ^ab^	0.71 ± 0.02 ^de^	8.91 ± 0.09 ^cd^
BF3	89.12 ± 0.16 ^a^	0.76 ± 0.03 ^d^	8.79 ± 0.08 ^d^
BF4	89.51 ± 0.12 ^a^	0.76 ± 0.01 ^d^	8.76 ± 0.11 ^de^
WBN1	83.11 ± 0.03 ^e^	1.60 ± 0.03 ^b^	9.52 ± 0.12 ^b^
WBN2	84.52 ± 0.07 ^c^	1.64 ± 0.02 ^a^	9.68 ± 0.05 ^b^
WBN3	84.11 ± 0.09 ^d^	1.63 ± 0.01 ^ab^	9.86 ± 0.04 ^a^
WBN4	84.47 ± 0.10 ^c^	1.55 ± 0.03 ^c^	9.53 ± 0.10 ^b^

Different small letter superscripts in the same column indicate that the values are significantly different at *p* < 0.05. BF1: coarse buckwheat flour. BF2–4: jet-milled buckwheat flour prepared by adjusting the feeding speed (145 r/min, 200 r/min and 250 r/min), respectively. WBN1–4: extruded whole buckwheat noodles prepared by BF1–4, respectively.

**Table 5 foods-11-02722-t005:** The analysis of cooking properties of WBN.

Sample	Water Absorption (%)	Cooking Loss (%)	Broken Rate (%)	Cooking Time (min)
WBN1	182.97 ± 5.23 ^a^	9.31 ± 0.12 ^a^	10.00 ± 1.92 ^b^	22.21 ± 0.23 ^b^
WBN2	154.21 ± 4.67 ^c^	7.86 ± 0.24 ^c^	6.67 ± 1.56 ^d^	23.29 ± 0.14 ^a^
WBN3	169.07 ± 1.73 ^b^	8.42 ± 0.16 ^b^	8.89 ± 1.56 ^c^	22.45 ± 0.15 ^b^
WBN4	184.19 ± 2.71 ^a^	9.47 ± 1.78 ^a^	12.22 ± 1.92 ^a^	21.12 ± 0.11 ^c^

Different small letter superscripts in the same column indicate that the values are significantly different at *p* < 0.05.

**Table 6 foods-11-02722-t006:** The analysis of textural properties of WBN.

Sample	Hardness (g)	Springiness	Cohesiveness	Chewiness (g)	Resilience
WBN1	3113.5 ± 43.12 ^c^	0.93 ± 0.01 ^b^	0.51 ± 0.01 ^c^	1478.73 ± 21.05 ^c^	0.23 ± 0.01 ^b^
WBN2	3348.41 ± 47.17 ^a^	0.95 ± 0.01 ^a^	0.57 ± 0.02 ^a^	1818.16 ± 19.01 ^a^	0.25 ± 0.01 ^a^
WBN3	3213.18 ± 51.70 ^b^	0.94 ± 0.01 ^ab^	0.54 ± 0.01 ^b^	1636.01 ± 27.18 ^b^	0.24 ± 0.00 ^ab^
WBN4	3079.13 ± 39.31 ^d^	0.93 ± 0.01 ^b^	0.51 ± 0.01 ^c^	1463.43 ± 19.04 ^c^	0.23 ± 0.00 ^b^

Different small letter superscripts in the same column are significantly different at *p* < 0.05.

## Data Availability

Data are contained within the article.

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
