# Peer review of "Effects of Jet Milling on the Physicochemical Properties of Buckwheat Flour and the Quality Characteristics of Extruded Whole Buckwheat Noodles"

_foods, 2022, doi:10.3390/foods11182722_

Round 1

Reviewer 1 Report

I have gone through the manuscript ID: Foods-1875743, titled “Effects of jet milling on the physicochemical properties of buckwheat flours and quality characteristics of extruded whole buckwheat noodles”. The research is interesting and has been well planned and executed. The results have been presented well and have been thoroughly supported with relevant discussion. The detailed report has been given below.

1.The research focused on processing technique, jet milling and its effect on the final extruded products, noodles made with whole buckwheat which might be a promising technique for developing gluten free noodles.

2.It is relevant and interesting, and buckwheat has been widely used for replacing refined wheat flour from noodles, bread, pasta, etc. and the major drawback was its coarse structure which has been addressed in this research.

3.Most of the studies have focused on ingredients and their effect on replacing the base material with buckwheat but here they have proposed an alternative processing technique “jet milling” to reduce the particle size of buckwheat flour which in turn improved its physico-chemical, rheological, colour and textural properties which evinced positive responses.

4.The paper is well written. The text is clear and easy to read.

5.The conclusions are consistent with the evidence and arguments presented.

6.They address the main question posed.The jet milling technique for preparation of buckwheat flour is very promising and would be helpful for preparation of other gluten free products also.

Thank You

Author Response

Thank you very much for your comments that has inspired and encouraged us.

We have made changes in the article and shown them in different colours.

Reviewer 2 Report

This study has investigated the effects of jet milling on the physicochemical properties of buckwheat flours and extruded buckwheat noodles. Several tests have been performed on the flour (particle size, color, hydration, pasting, rheological and textural properties) and noodles (SEM, cooking properties, texture and fiber content). The manuscript is well structured and has reported valuable results. It need to revise.

Comments:

Lines 85, 103, 129, 163, 175: Use a number for the citation of “AACC” in the text and add the reference to the reference list.

Line 100: Explain more and add a reference for particle size measurement.

Line 131: How did you determine the strain?

Line 143: please add the name of textural test (e.g. TPA)

Line 169: How did you determine the optimal cooking time?

Line 175: Add the reference AOAC to the reference list.

Line 182: Add the country and the company.

Figures and tables should be placed at the end on each section

Line 203: Add the SEM images to the main document.

Figure 2: Add error bars

Line 402: please explain more, add some references and compare the obtained results with other studies.

Conclusion: Please specify which treatment was the best 

Author Response

1.Lines 85, 103, 129, 163, 175: Use a number for the citation of “AACC” in the text and add the reference to the reference list.

Thank you for the comments. We have made the changes.

2.Line 100: Explain more and add a reference for particle size measurement.

Thank you for the comments. We have made the changes in Line 99-101.

“The particle size distribution of samples was tested in wet method mode by a S3500 Particle Size Analyzer (Microtrac Co.Ltd, USA). Referred to Yu et al. [11], ultra-pure water was used as dispersant, and the refractive index of the samples was set to 1.434. The data was analyzed by system software FLEX 10.5.3.”

3.Line 131: How did you determine the strain?

Thank you for the comments.

Based on previous studies, stress sweep tests at 1.0 Hz were firstly conducted in order to define the linear viscoelasticity zone in the range of strain from 0.01% to 1%. Then we set the strain at 0.5% for dynamic rheological test. The experimental results showed that when the strain was 0.5%, the response of samples was significant enough in the linear viscoelastic region.

4.Line 143: please add the name of textural test (e.g. TPA)

Thank you for the comments. We have made the changes.

5.Line 169: How did you determine the optimal cooking time?

Thank you for the comments.

Referred to AACC Method 66-50 which has been shown in 2.12 (Line 166-169), we determine the optimal cooking time.

Specifically, 10 g of noodles were placed into 500 ml of boiling distilled water and cooked at the optimal cooking time, as judged by slightly squeezing between two-glass slides till the white inner core disappeared.

6.Line 175: Add the reference AOAC to the reference list.

Thank you for the comments. We have made the changes.

7.Line 182: Add the country and the company.

Thank you for the comments. We have made the changes.

8.Figures and tables should be placed at the end on each section

Thank you for the comments. We have made the changes.

9.Line 203: Add the SEM images to the main document.

Thank you for the comments. We have made the changes.

10.Figure 2: Add error bars

Thank you for the comments. We have made the changes.

11.Line 402: please explain more, add some references and compare the obtained results with other studies.

Thank you for the comments. We have made the changes.

“The textural characteristics of noodle product play a crucial role in consumer acceptance. What is more, hardness is the most important index of cooked noodles’ quality. Noodle textural attributes were revealed in Table 6. As listed in Table 6, the textural parameters of cooked buckwheat noodles were strongly linked with the textural parameters of gel, of which the WBN2 exhibited significantly higher hardness, springiness, cohesiveness, chewiness, and resilience. These results were contributed by stronger gel network structure due to fine particle size and moderate starch damage. Some studies [36,37] have suggested that, to a certain extent, the finer the particles, the greater the texture parameters such as hardness and elasticity of the noodles, which was conducive to improving the consumer acceptability of products.”

12.Conclusion: Please specify which treatment was the best

Thank you for the comments. We have made the changes in Line 450-452.

“The physicochemical properties, cooking properties, and microstructure of extruded noodles were improved significantly by the proper grinding process where the feeding speed of jet milling was 145 r/min and the particle size (D50) of flour was reduced to 65.86 μm.”

Reviewer 3 Report

The manuscript was to investigate the effects of jet milling on physicochemical properties of buckwheat flours and quality of extruded whole buckwheat noodles.

1. The topic is not new, so the authors should emphasize the novelty of the study. For example, Did you have any potential application in food industry? 2. In the methods, please provide the detail of some key parameters such as how to determine the damage starch.

3. Moreover, in the discussion and results, How did you explain the effects of jet milling and extrusion on dietary fiber content? It looks no significant changes before and/or after extrusion.

4. The authors should re-organize the discussion section to emphasize the importance of the study.

Author Response

1.The topic is not new, so the authors should emphasize the novelty of the study. For example, Did you have any potential application in food industry?

Thank you for the comments.

The research focused on processing technique, jet milling and its effect on the final extruded products, noodles made with whole buckwheat which might be a promising technique for developing gluten free noodles.Most of the studies have focused on ingredients and their effect on replacing the base material with buckwheat but here we have proposed an alternative processing technique “jet milling” to reduce the particle size of buckwheat flour which in turn improved its physico-chemical, rheological, colour and textural properties which showed positive responses. The relationship between particle size and properties of extruded buckwheat noodles were not reported before. The results revealed that moderately ground buckwheat jet-milled powder led to the highest pasting viscosity, gel properties, lightness, best cooking quality and texture properties of extruded noodles. Moreover, jet milling increased soluble dietary fiber (SDF) content and SDF got further increased after extrusion, which was beneficial for the application of buckwheat flour in food processing.

2.In the methods, please provide the detail of some key parameters such as how to determine the damage starch.

Thank you for the comments. We have made the changes in Line 100-109.  

3.Moreover, in the discussion and results, How did you explain the effects of jet milling and extrusion on dietary fiber content? It looks no significant changes before and/or after extrusion.

Thank you for the comments.

The lowercase letters in the previous figure only showed the significant differences of dietary fiber content due to jet milling before or after extrusion, but not before and after extrusion.

Before extrusion, jet milling increased SDF content and reduced IDF content gradually, nevertheless TDF content was almost unchanged compared to BF1, suggesting that jet milling gave rise to a redistribution of fiber components in TDF. This might be caused by mechanical shear during jet milling process, which led to the fracture of some bonds of IDF and its conversion into soluble polymer.

After extrusion, SDF got further increased and IDF decreased. The TDF was also increased after extrusion significantly. The reason was that during the extrusion process, glycosidic bonds of insoluble polysaccharide molecules were broke and insoluble polysaccharide molecules were converted into smaller soluble constituents. Some studies have shown the similar results, that the extrusion processing had a positive influence on total and soluble dietary fiber content.

4.The authors should re-organize the discussion section to emphasize the importance of the study.

Thank you for the comments. We have made some necessary changes.